# TS-BERT: A fusion model for Pre-training Time Series-Text Representations

## Abstract

There are many tasks to use news text information and stock data to predict the crisis. In the existing research, the two usually play one master and one follower in the prediction task. Use one of the news text and the stock data as the primary information source for the prediction task and the other as the auxiliary information source. This paper proposes a fusion model for pre-training time series-Text representations, in which news text and stock data have the same status and are treated as two different modes to describe crises. Our model has achieved the best results in the task of predicting financial crises.

## 1 Introduction

Forecasting the stock market trend from multiple angles is of great significance to economics researchers and sociological researchers.

First of all, there are extensive studies in finance to solve the problem of forecasting the stock market. There are two main motives for them. The first is that predicting the stock market trend could reveal the specific factors that drive the stock market trend. Second, by predicting the stock market trend, it is possible to evaluate the pricing of particular assets Pönkä (2017).

The influence from the stock market would spread to other areas due to the significant impact on the society from financial. How to predict and prevent significant stock volatility crises is a critical research topic. Economic researchers in academia and the financial industry have never stopped their research in this area. Related research methods are constantly updated.

Financial time-series forecasting is one of the most important research tasks in these fields. Economics researchers are increasingly accepting the use of deep learning for related research. More and more tasks use AI technologies such as machine learning and deep learning to predict financial time series. Researchers in related fields have created various models and published a large number of papers accordingly. After introducing deep learning models into this field, the results obtained are significantly better than traditional machine learning models Sezer et al. (2020). For example, the time series forecasting model based on the transformer Wu et al. (2020a), and a time series forecasting model based on graph neural network Wu et al. (2020b). Researchers are becoming more and more interested in developing financial time series forecasting models Thakkar & Chaudhari (2021). While the modelling methods are becoming more and more diversified, the complexity of the model is also increasing. Fortunately, these behaviour results are gratifying, and the accuracy of financial time series forecasting tasks is constantly improving.

Not only is the model itself becoming more powerful and more complex, but even the information on which the prediction task itself is based is also becoming more and more diverse. From time-series data itself to macro-financial factors to text data such as news, the data of financial time series forecasting tasks have become more abundant. Especially for text data, text mining has now become a significant research field. It has been applied in many fields such as medical treatment, business, finance, education, and many others Elagamy et al. (2018).

News content is one of the most critical factors affecting the stock market. Analysis of the impact of news when predicting stock market volatility would lead to more reasonable forecasts. Stocks are one of the most important components of financial markets, and they could directly impact regional and global economies. Precise forecasts of stock trends could help support critical business and financial decisions and even political decisions.

Therefore, we use stock time series data and news text data to enter the main methods of this article on stock market trends. The primary method of our project is to improve the accuracy of prediction based on the fusion representation of text and stock time-series data.

## 2 Related work

The use of data mining techniques to analyze and predict the stock market has been widely recognized Nikfarjam et al. (2010). This method mainly uses structured data, such as past prices, historical returns, or other stock characteristic indicators. The task of using data mining to predict financial time series has achieved many results. Samitas et al. (2020) has established an early warning system, which dramatically improves financial risk early-warning accuracy and could give early warning signals in advance.

The method of text mining is different from that of data mining. It is not easy to extract relevant information from unstructured data such as news. So this type of method has relatively few applications in financial time series forecasting tasks.

The trend of stocks is entirely random. Using data mining methods to predict the trend of stocks is very dependent on a large amount of data. However, historical data that stretches the time axis too long does not have a high reference value for the current state Patel et al. (2015). Furthermore, they believe that it is not easy to provide a reasonable explanation effect simply by using numerical prediction Schumaker et al. (2012). News content is one of the most critical factors affecting the stock. If the news is classified in advance, and then these tags are passed in when forecasting, better results could be obtained Nikfarjam et al. (2010).

News text data usually contains many events, and these events could be used for the supplementary analysis of random behaviour. There is a lot of free news and financial data. If both could be used at the same time, it may improve the interpretability of the predictive model. Moreover, they proved that the correct available information could improve the performance of sequence prediction models.

Text information has a very unfriendly feature: their language structure and writing style are very different. It is not easy for machine deep learning technology to learn information and patterns from such unstructured data. This difficulty is mainly reflected in the modelling and training phases. In order to solve these problems, there are two leading solutions: subject terms, modelling, and keyword modelling.

The solution we currently use is keyword modelling, which is used to characterize news text.

Many researchers use news in stock forecasting tasks. For example, the news labels are input into the predictor as a feature of stocks Nikfarjam et al. (2010), and Fung et al. (2002) also proposed a system that only uses the news to predict stock trends. A stock trading early warning system is also established through the classification of news Mittermayer & Knolmayer (2006). News is used in such tasks because news has an impact not only on the stock market itself but also on the stock market participants. They discover that specific news articles can influence investors' decision-making Li et al. (2014). In addition, they prove that the text information of the news could help investment institutions conduct intraday risk management Groth & Muntermann (2011).

There are many examples of using news in tasks related to stock forecasting, and most of them have achieved better results. However, their method is to classify the news text as a stock feature or only use the news text without using stock data.

In addition to fusing multiple information together, some researchers use different models and different types of information to make different predictions He et al. (2019). Then they

merge these single models and use a model fusion solution to improve the prediction effect. Model fusion could perform various operations to fuse two or more models and multiple specific features. It must be noted that these operations may be different from the possible operations performed by feature fusion. Such operations and optimization could be applied to a wide range of models. Extensive experiments have proved that the performance of the fusion model is 3%-5% higher than the performance of a single model.

Fusion models usually require the use of information features from multiple sources, and then use the superior performance of different models for certain types of information to improve the final performanceThakkar & Chaudhari (2021).

Since some studies already prove that news text alone could also effectively predict the stock market, we suppose that news could be an independent observational state of the stock market. Using news text and stock price data simultaneously and treating them as two modalities, we could use a multi-modal learning method in the prediction task.

## 3 TS-BERT

### 3.1 The BERT model

BERT Devlin et al. (2019) proposes to learn language representations by using a "masked language model" training objective. In more detail, let $x = x_1, ..., x_L$ be a set of discrete tokens, $x_l \in X$. We can define a joint probability distribution over this set as follows:

$$p(x \mid \theta) = \frac{1}{Z(\theta)} \prod_{l=1}^{L} \phi_l(x \mid \theta) \propto \exp\left(\sum_{l=1}^{L} \log \phi_l(x \mid \theta)\right)$$

Use BERT to establish a self-supervised key semantic feature extraction method, and achieved the best results. This process avoids the time for manual tagging of keywords and does not require any prior knowledge. This allows us to easily apply this method to huge financial news data sets.

### 3.2 TS-BERT

Our world experience is multi-modality we see objects, hear sounds, feel textures, smell smells, taste tastes. Modality refers to the way something happens or experiences. When a research question contains multiple modalities, it has the characteristics of multi-modality. In order for artificial intelligence to make progress in understanding the world around us, it needs to interpret these multi-modal signals simultaneously.

For example, images are usually associated with tags and text explanations. The text contains images to more clearly express the article's central idea. Different modes have very different statistical properties.

In our task, the volatility of stocks is our goal. The statistical characteristics of stock volatility exist in the time series data of stocks and the news text related to stocks.

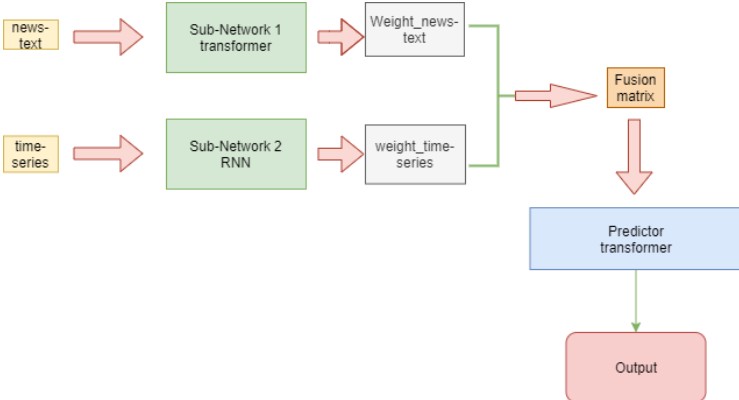

Figure 1: Full-model Diagram.

### 3.2.1 Fusion Embedding

The fusion embedding method could be divided into two parts. The first part is to embed the news text, the second part is to embed the stock data, and then the third step is to fuse the two. In terms of design, this is a multi-modal neural network learning method.

### 3.2.2 News Embedding

In text embedding, we use keyword extraction technology, which uses BERT to embed to create keywords and key phrases that are most similar to news text.

Although there are many methods available for keyword generation, such as Rake, Yake, and TF-IDF, the performance of these once-popular methods is no longer so powerful in the current deep learning field. Due to the advent of pre-trained models, it is easy to create a more powerful method to extract keywords and key phrases. This is the design basis of our method, using Bert embedding and superficial cosine similarity to find the most similar sub-phrases in the news text to the news itself.

First, use BERT to extract news embedding to obtain news-level embedding. Then use no-gram to extract the embedding of the word. Finally, we use cosine similarity to find the most similar words or phrases to the news. Then we define the most similar words or phrases as the words and phrases that best describe the exclusive news.

We feed the news to BERT, and obtain the contextual feature vector of each word. The vectors of words in a news is averaged in order to get its news embedding vector. Then we choose the words close to news embedding vector. The idea is that a keyword should capture meaning of news, and thus should be closer to the news embedding. The similarity of the embeddings to the news embedding is obtained using cosine similarity metric.

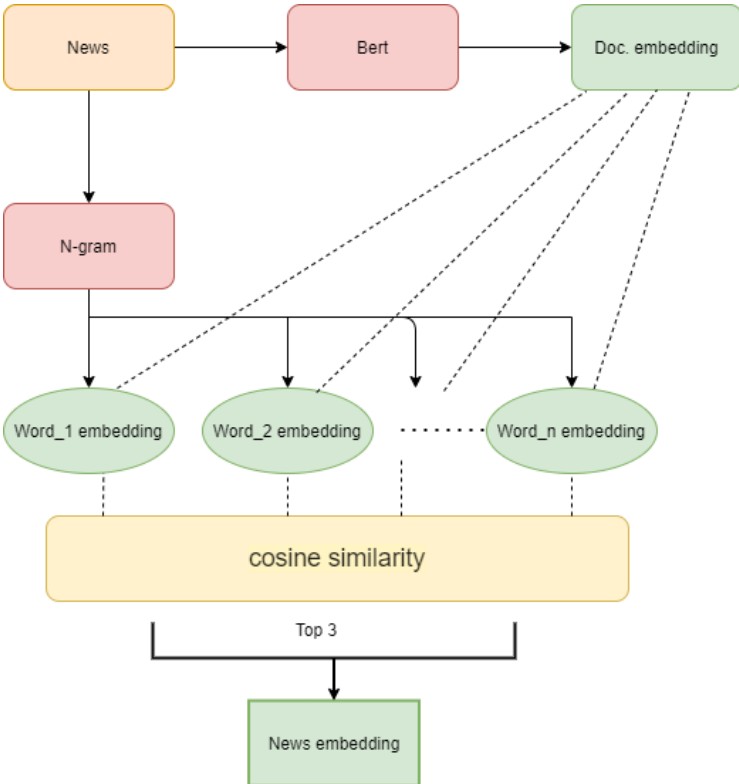

Figure 2: News embedding.

The advantage of this method is that we do not need to start training from scratch and adjust the parameters multiple times. The performance of BERT itself has been widely recognized.

### 3.2.3 Stock time series data Embedding

There are also many modelling methods for stock time-series data. However, because we do not need to use a very complex model in this task, we choose to use a basic RNN to embed the stock time series data.

Embedding connection Although combining different modalities or information types to improve the effect is intuitively an attractive task, how to combine different noise levels and conflicts between different modules in practice is a challenge. The most common method in practice is to connect high-level embeddings of different inputs and then apply softmax.

The problem with this approach is that it would give equal importance to all sub-networks/modes, which is very unlikely in reality. News and stock prices may not have the same influence on the forecast, so we need to calculate the different weights they play.

### 3.2.4 Weighted combination of networks

Because the relationship between the two modalities of text features and time-series features is relatively complicated, we could not only use simple fusion methods to describe this relationship. And it would cause the fusion of feature vectors to have too high dimensionality.

We use MCB to solve this problem. MCB pools the fused features. First, the vector outer product calculation is performed on the feature vector of each location of the feature map obtained by the convolution. Then sum pooling is performed on the results of the outer product calculation of all locations to obtain the feature vector x. x gets the final feature after signed square root and L2 normalization.

The operation of bilinear pooling can be expressed as:

$$B(X) = \sum_{s \in S} X_s X_s^T$$

MCB maps the result of the outer product to a low-dimensional space, and does not need to explicitly calculate the outer product.

## 4 EXPERIMENT

### 4.1 Data

We used data from the kaggle competition. We carry out some necessary processing of these data. In the end, the data set we used for the experiment contained data from a total of 300 companies from NYSE and NASDAQ. The period is from 2008 to 2021, with a total of more than 2,800 trading days. Each day's data for each company includes the corresponding stock price and news related to the company.

We consider the data of 300 companies as a whole, so our data set length is around 840,000. We split the data set into two parts: training and testing, 85% is the training set, and 15% is the test set. In addition, we regard the top 5% and tail 5% of the stock's daily rise and fall in a cycle as crisis and the rest as crisis-free.

### 4.2 Baseline Methods

For contrastive experiments, we consider baselines with two different methods:GCN classifier and CNN classifier with LSTM embeddings.Since these models do not have multi-modal application cases similar to our text-time series data, this result could not be considered as the best performance of the model on such tasks.

#### 4.2.1 GCN classifier

We use words and stock features as news nodes to construct news with a graph structure. Then use TF-IDF to characterize the relationship between words and news. In the training phase, we use a fixed-length window to divide the crisis or not as a monitoring label for news.

#### 4.2.2 CNN classifier

We use LSTM to extract the key semantics of the text, and then use CNN to extract the key features of the semantics.It should be noted that only news data is used in CNN classifier.

### 4.3 Results

As shown in table1, both our model and baseline model have high precision, but fewer false warnings are crucial for crisis prediction tasks, so we also calculated the recall.

Table 1: Result of precision and recall.

| Model | Precision | Recall | F1 Score |
|---|---|---|---|
| TS-BERT | 0.98 | 0.95 | 0.96 |
| MTGNN | 0.92 | 0.88 | 0.90 |
| CNN | 0.91 | 0.76 | 0.83 |

### 4.3.1 Ablation experiment

In order to verify that the text and time series data respectively made positive contributions to the experiment, we did an ablation experiment and showed the results in table2

Table 2: Comparison of results between single modal experiment and dual modal experiment.

| Model | Precision | Recall | F1 Score |
|---|---|---|---|
| TS-BERT(news only) | 0.84 | 0.89 | 0.86 |
| TS-BERT(stock only) | 0.94 | 0.81 | 0.87 |
| TS-BERT | 0.98 | 0.95 | 0.96 |
| MTGNN | 0.92 | 0.88 | 0.90 |
| CNN | 0.91 | 0.76 | 0.83 |

## 5 CONCLUSION

Using multi-modal learning methods to improve the accuracy of stock risk volatility prediction is one of the main directions of our work. In future work, we have additional possible improvements.

At present, we use the top3 self-relevance keywords in the news as news representations, and there may be possible that the impact of these three keywords on the stock market is not the same. We could score the relevance of the news stock classification before extracting the keywords. Then we use the highest score as the stock classification tendency of the news and then extract the top3 words of this classification tendency in the news.

Another possible improvement is that we may design a new crisis modelling method, which aims to accurately describe the crisis fluctuations in the stock market and improve the interpretability. And then, using a more complex and more powerful time-series data embedding method may also improve the performance of our model.

In order to be more in line with the actual stock market, perhaps the weight ratio of precision to recall is not the same as in the standard F1. Therefore, changing the recall weight is also one of the important applications of the F1 score. After the change, the formula of the F1 score could turn to:

$$F_\beta = (1 + \beta^2) * \frac{Precision * Recall}{(\beta^2 * Precision) + Recall}$$

If necessary, experimental data, model structure, etc., may be further improved.

### Acknowledgments

Use unnumbered third level headings for the acknowledgments. All acknowledgments, including those to funding agencies, go at the end of the paper.

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
