# OpenReview forum: "TS-BERT: A fusion model for Pre-trainning Time Series-Text Representations"
_ICLR.cc/2022/Conference — ICLR 2022 Submitted_

### Official Review · Reviewer_7P8j · 2021-10-28

**Correctness:** 2
**Technical Novelty And Significance:** 1
**Empirical Novelty And Significance:** 1
**Recommendation:** 3
**Confidence:** 5

**Main Review:**

Reasons to accept:

-

Reasons to reject:

I appreciate the authors’ effort. I am certain that they have made a great effort to develop the model and prepare the paper. I’d like to also remind that my task, as a reviewer, is to provide a helpful review so that the authors can receive an outsider view.

- In my opinion this article is more like a blog post which is *badly* formatted in a scientific paper. More specifically: The language is below standard, the related work coverage is very limited, the claims are contradictory, the model is too naive, the experiments are insufficient, the baselines are weak, and the analysis is superficial.

- The paper has no contribution in my opinion. The main idea of the paper has been proposed before [1], and is not cited in the paper.

- The authors neither in the introduction nor in the related work mention their actual contribution compared to existing works. The reader should infer this himself.

- There are a number of contradictory claims in the paper. The authors state:
“The trend of stocks is entirely random.”
If it is random, then why would you make an attempt to predict it?
Again the authors state that their baselines are weak:
“Since these models do not have multi-modal application cases similar to our text-time series data, this result could not be considered as the best performance of the model on such tasks.”
If they are weak, why you didn’t use better baselines?

- BERT already has a document vector representation called [CLS]. There is no need to take average of words.


[1] Fune et al., Stock prediction: Integrating text mining approach using real-time news. CIFEr 2003.

**Summary Of The Paper:**

The authors propose a model for stock prediction by aggregating the embeddings of automatically mined news keywords and the embeddings of stock series. The authors evaluate their model in a large dataset and show that it outperforms a TF-IDF based model and a CNN based model.

**Summary Of The Review:**

The presentation of the paper is below standard, it has no contributions, and the experimental setup is not convincing.

---

### Official Review · Reviewer_oxEh · 2021-11-03

**Correctness:** 2
**Technical Novelty And Significance:** 1
**Empirical Novelty And Significance:** 2
**Recommendation:** 1
**Confidence:** 4

**Main Review:**

Strengths:
- The method is well-motivated and the problem is interesting.
- The proposed method achieves the best performance in the empirical study,

Weaknesses:
- There are too many apparent and fundamental mistakes in this paper:
    - Though it looks similar, this paper does not follow ICLR's official format. For example, the title of each subsection is in lower case and the citations are in the wrong format.
    - There are some outstanding drafts at the end of the paper (see Page 7 before the acknowledge section).
    - The authors did not hide the acknowledge section in this submission, though there is no information leak.
    - There are grammatical errors in most of the paragraphs.
- The technical contribution and novelty are relatively limited or not well presented.
- Minor problems/suggestions:
    - Please add a link or reference to the Kaggle dataset you use.
    - What does MCB mean. Please cite the paper which proposes it or briefly explain it.
    - This picture is so unclear. It would be much better if using a PDF version.


**Summary Of The Paper:**

This paper introduces a method that combines the representation of text and time series to make financial crisis predictions. The proposed methods outperform all the baseline methods.

**Summary Of The Review:**

This paper proposes a new and powerful method to solve an intersting problem. However, the technical contribution is limited and the presentation quality needs to be greatly improved to meet the bar of ICLR.

---

### Official Review · Reviewer_YK4B · 2021-11-03

**Correctness:** 3
**Technical Novelty And Significance:** 2
**Empirical Novelty And Significance:** 2
**Recommendation:** 3
**Confidence:** 4

**Main Review:**

Strong points:

- The combination of text and financial data is a worthy area of study (although this work is not the first to try to do this).  Using BERT is a promising approach here.

- An ablation study showed that both the news and stock portions of the model were important to get the best performance.

Weak points:

- The proposed model is not particularly novel, as far as it is explained.  Surely the authors are not the first to combine text and financial data, and the precise novelty is not well argued.

- The fusion aspect of the approach is fairly pedestrian and does not consider how text and stock prices interact and change over time.

- The details of the proposed TS-BERT model are a bit unclear in the paper. Figure 2 is helpful but it does not suffice. There needs to be equations in the paper describing the precise details of the model. I definitely could not replicate the authors' method based on what is included in the paper.

- It is unclear why keyword extraction was performed based on the BERT embeddings, instead of simply using the BERT embeddings.  There would be a loss of information in this step.  If this step is useful, experiments should be provided which show it. Similarly, the role of N-grams in Figure 2 is unclear.

- The training algorithm for the model needs more explanation.  What is the loss function, what is the optimization algorithm, and how is fine-tuning used?

- Only one dataset is used in the experiments.

- There are insufficient baselines.  Other state of the art text + stock price models, and published models considering these aspects separately, should be compared to (e.g. models which were mentioned in the related work section).

- Both of baselines which were used are a bit strange and unclearly explained.  The GCN method represents word and stock features as a graph, but this data is not naturally graph structured, and it is unclear how the graph was constructed or what the rationale was.  The use of TF-IDF in this context is also very unclear.  Similarly, it is not clear why the CNN model also includes an LSTM component, or how these two were put together.  I don't know why there weren't simply separate LSTM and CNN baselines.

- There are numerous grammatical errors throughout the manuscript.  There are other clarity and framing issues as well, e.g. the paper goes back and forth between saying that the task is stock forecasting (i.e. predicting stock price), and predicting the occurrence of crises.

Additional feedback / minor suggestions:

- The equation regarding BERT doesn't seem to be the right one. Or at best, it does not capture the unique aspects of BERT, such as the transformer-based architecture, bidirectional representations, and the masked language model and next sentence prediction training objectives.

- There needs to be a citation for the MCB method. What does the acronym stand for? Including equations to explain it would help as well, if space allows.

- "table1", "table2" - should be "Table 1", "Table 2".

- The Acknowledgments stub from the conference template was left in the paper.

- The title is not consistent in its use of capitalization.

- BERT is not consistently capitalized throughout.

- Section 3.2.3 heading has inconsistent capitalization. Below it, "Embedding connection" looks like it should be a heading, instead of the first two words of a different sentence.

- Many citations should be put inside parentheses, especially when they occur at the end of sentences. E.g. "Sezer et al. (2020)" should be "(Sezer et al., 2020)" in its use on page 1. Use the Natbib \citep{} command.

- Is MTGNN the method which is referred to as GCN in Section 4.2? Name it consistently.

**Summary Of The Paper:**

The paper proposes a method for predicting stock market crises using a deep learning approach which combines time series stock market data with text from news articles.  Experiments find that the method works better than the same model using only news or only stock prices, and a couple of deep learning baselines.

**Summary Of The Review:**

While the combination of text and financial stock data via BERT is an interesting task, the paper presently has multiple serious issues in novelty, clarity, and experimental rigor.

---

### Decision · Program_Chairs · 2022-01-20

**Decision:**

Reject

**Comment:**

The paper proposes a method for predicting stock market crises using a deep learning approach which combines time series stock market data with text from news articles. Their experiments show that the proposed method works better than the same model using only news or only stock price data, and a couple of deep learning baselines. All the reviewers pointed out that this paper is lack of novelty and significant technical contributions. The experiments are performed on a single dataset with incomplete baselines, and hence insufficient to support the claimed advantages of the proposed method. The writing quality is not up to the standards of an ICLR papers, with too many grammatical mistakes, typos, and unjustified arguments/claims.  The clarity of the writing is poor.

The authors did not provide their rebuttal.